# Self-Assessment of the Pelvic Floor by Women Practicing Recreational Horseback Riding

**DOI:** 10.3390/ijerph19042108

**Published:** 2022-02-13

**Authors:** Monika Urbowicz, Mariola Saulicz, Aleksandra Saulicz, Edward Saulicz

**Affiliations:** 1Physiotherapy Practice “Physiotherapy—A Strong Foundation”, 43-190 Mikołów, Poland; monika.urbowicz@gmail.com; 2Institute of Physiotherapy and Health Sciences, Jerzy Kukuczka Academy of Physical Education, 40-065 Katowice, Poland; m.saulicz@awf.katowice.pl (M.S.); e.saulicz@awf.katowice.pl (E.S.); 3School of Public Health & Social Work, Queensland University of Technology, Brisbane, QLD 4059, Australia

**Keywords:** women, horseback riding, pelvic floor, bladder function, bowel function, prolapse symptoms, sexual function

## Abstract

The aim of this study was to compare the condition of the pelvic floor in women who are involved in regular recreational horseback riding, with both physically active women as well as women not undertaking any recreational physical activity. Taking into account horseback riding and physical activity, 140 healthy women aged 17 to 61 were divided into three groups: women practicing horseback riding (WPHR) (46 persons), physically active women (PAW) (47 persons) and women not physically active (WNPA) (47 persons). The Australian Pelvic Floor Questionnaire (APFQ) was used to measure the extent of pelvic floor dysfunctions in women from all three groups. The lowest average values were found in the group of women practicing recreational horseback riding, and the highest in the group of women not physically active (95% CI: 0.61–1.15 vs. 0.87–1.44 —bladder scores; 0.82–1.32 vs. 1.24–1.8—bowel scores; 0.07–0.33 vs. 0.08–0.35—prolapse of reproductive organs scores; 0.4–1.07 vs. 0.49–1.3—sexual function). Statistically significant intergroup differences were recorded only for the bowel function rate (*p* = 0.021). The overall pelvic floor dysfunction rate in the WPHR group was lower when compared with both control groups (95% CI: 2.15–3.62 vs. 2.34–3.54 in women from PAW group and vs. 3.0–4.56 in women from WNPA group). Based on this study, it can be concluded that all of the pelvic floor related symptoms, their frequency, and severity levels do not qualify recreational horseback riding as being a risk factor for developing pelvic floor dysfunction in women.

## 1. Introduction

In recent years, there has been an increasing interest in horseback riding as a form of leisure and recreational activity for women in Poland. According to the data by the Polish Equestrian Federation [1], the number of registered equestrian clubs in Poland increased from 420 in 2014 to 528 in 2019. In addition, the PEF development strategy of Polish equestrianism 2019–2022 [1] states that the actual number of equestrian clubs in Poland is at least twice as high. This data corresponds with the data from the Polish Horse Breeders Association, estimating that the 2019 population of noble horses and ponies was at 140,000 heads [2]. In 2014, recreational equestrian activities were estimated to be practiced by 95,000 people [3], which, based on the growth dynamics of equestrian centers, could have increased by 20% by 2020. Equestrian activities such as sport horseback riding in Poland are heavily feminized. According to the Polish Equestrian Federation, 84% of the competitors in all equestrian disciplines are women [4].

Horseback riding is a physical activity of a moderate effort at 3–6 MET, with the aerobic cost increasing as the horse speed accelerates [5]. Positive effects on back pain were indicated among the physical aspects of recreational horseback riding, and this fact was partly confirmed by a study with a use of a horseback riding simulator [6]. The literature review of the health effects of hippotherapy shows that most of the reports confirm the positive impact on the so-called gross motor skills, muscle tension, posture, and balance of the rider [7]. It has also been demonstrated that horseback riding lowers cortisol levels in adults as well as increases their serotonin levels and cerebral alpha waves. Additionally, it improves self-evaluation of their quality of life and increases eagerness to practice sports [8,9].

Positive effects of horseback riding on body posture and motor skills are connected with movements of the back of the horse that forces rider’s pelvis to move. Depending on the speed of riding, these movements may almost accurately imitate pelvis movements when walking. During the slowest horse gait, the walk, the rider’s body is subjected to a rocking motion of up-down, forward-back, and right-left movements. Hence, horse movement is described as three-dimensional. When the saddle is raised, the pelvis is subjected to a retroversion, and when it falls the pelvis undergoes an anteversion, which is opposite to the direction of the saddle movement. The forward-back movement of the pelvis is superimposed by up-down and right-left movements, which in practice looks as if the ilia were moving along two ellipses, separate for the right side and the left side [10]. During the trot, which, just like the walk, is symmetrical, there is a suspension phase when all of the four limbs of the horse are above the ground. The entire horse rises and falls, and a stronger up-and-down action occurs. The trot requires the rider to adapt to the forces generated by the horse in the vertical axis so as not to bounce off the saddle, but rather follow the movement of the horse back harmoniously [10]. The fastest horse gait used for dressage, show-jumping, and recreational riding is the canter, which is an unsymmetrical gait, where the leading limbs are on the same side of the horse. The canter has a long suspension phase with the back of the horse swinging between the “uphill” and the horizontal position. During the canter, the rider is inclined slightly forward, and the asymmetric movement of the horse makes their pelvis continue rotating around the ellipse, but their whole body remains in the fencing posture (i.e., with their leg on the horse’s leading limb in a forward position and the leg on the opposite side as a trail leg). In the canter there are much more extensive yaw rotation movements of the rider’s trunk [10]. A female horseback rider distributes the weight of her body around the area of her ischiatic tubers and pubic symphysis (men do not lean against pubic symphysis due to the anatomical differences) and on the inner sides of her thighs, and to a lesser extent on her feet which are supported in the stirrups [11]. The horse rider’s trunk and pelvis movements as well as the horse movements under the rider have a direct impact on the points of the greatest pressure of the body on the horse’s back, such as the ischiatic tubers, pubic symphysis, and the area of the lesser trochanters. Between the pressure points and the saddle, the rider’s pelvis performs, among others, the roll movement in the opposite direction to the roll rotation of the horse’s back (and thus the saddle itself), the pelvis does not follow the full range of the horse’s movement. In some occasions the pelvis makes a less extensive movement than the saddle, and sometimes the movement of the pelvis is more extensive than the movement of the saddle—it depends on the type of the horse gait and the rider’s experience. In addition, less experienced riders find it difficult to follow the vertical oscillations in the trot, which may lead to the pelvis bouncing in the saddle [10]. Such movements with unfitted garments, excess movements, or poor coordination can cause friction and even injuries in the area of contact between the body and the saddle, and that is what the equestrian clothing industry is trying to prevent [12].

Thus, horseback riding is a form of physical activity that, more than any other physical activity, directly affects the female pelvis. This effect is both, movement-related (connected with the pelvic movements adjusted to the speed of riding), as well as sensation-related (resulting from a direct contact between the lower part of the pelvis and the saddle). Therefore, by engaging the pelvis to such an extent, horse-back riding has a significant impact on the female pelvic floor. During the vaginal electrode EMG tests, it was demonstrated that the tension of the female pelvic floor increased with the increase in the horse’s speed rate. The highest tension was recorded in the canter in a two-point position, slightly lower in the sit-position canter, and it returned to the resting state when the horse was stationary [13]. On the basis of these studies, a hypothesis was adopted that horseback riding has a positive effect on preventing pelvic floor dysfunction in women. Other studies showed that supporting the pelvic floor training with hippotherapy in women with stress incontinence increased the resting tension of the pelvic floor and contributed to a stronger tension during active pelvic floor muscle contraction [14]. The analysis of the effects of horseback riding on a sexual function and the condition of the lower urinary tract in women and men involved in equestrian activities, as opposed to cycling, did not reveal any negative effects on sexual functions or the condition of the lower urinary tract [15]. However, there are no reports in the literature with regard to the impact of recreational horseback riding by women on the condition of their pelvic floor, taking into account the presence of the symptoms of urinary, intestinal and pelvic organ prolapse as well as sexual dysfunction. Therefore, the aim of this study was to compare the condition of the pelvic floor in women who are involved in regular recreational horseback riding with both physically active women as well as women not undertaking any recreational physical activity. 

## 2. Materials and Methods

### 2.1. Study Design and Subjects

A cross-sectional study investigated the occurrence of symptoms of pelvic floor dysfunctions among 140 potentially healthy women aged 17 to 61. Taking into account the horseback riding and physical activity, the analyzed women were divided into three groups: women practicing horseback riding (WPHR), physically active women (PAW) and women not physically active (WNPA). 

The criterion for inclusion in the group of women engaged in recreational equestrian activities was to practice horseback riding for at least one year for a minimum of 1 h per week. The criterion for inclusion in the group of physically active women was to do other forms of recreational physical activity also for at least one year for a minimum of 1 h per week. Finally, the criterion for inclusion in the group of women not physically active was lack of doing any recreational activities. The exclusion criteria for all of the three groups were as follows: poor health, chronic cardiovascular and respiratory diseases, metabolic disorders, renal diseases and mobility impairment. Women who did not know the Polish language to the extent allowing them to understand the questionnaire and answer the questions in an informed manner were also excluded from the study. In the group of women practicing horseback riding, the additional criterion for exclusion was regular recreational exercise of additional physical activity and in the group of women physically active (i.e., practicing more than one form of physical activity). The enrolment process of the surveyed women is presented in Figure 1. 

Finally, 46 women aged between 17 and 61 were enrolled in the group of women systematically involved in a recreational horseback riding. In this group of women, the actual number of years of equestrian experience varied between 6 and 32 (16.1 ± 6.2) and the number of hours they spent on horseback riding was between 1 and 6 h a week (3.3 ± 1.9). A total of 47 women aged 21–60 were included in the group of physically active women, most of whom attended fitness activities (42.6%) for 1 h twice a week, did jogging (19.2% of the group) between 1 and 7 h a week or attended the gym (14.9%) one to three times a week. The group of women not physically active comprised of women aged 20–60 who declared a lack of any recreational form of physical activity taken regularly. 

### 2.2. Data Collection

The study was conducted using anonymous surveys. A group of women involved in recreational horseback riding was selected from all over Poland. The questionnaires were distributed during nationwide training on horseback riding, at the national conference of hippotherapists in Poznań, and among women who use the services of boarded horses in Silesia. Printed copies of anonymous surveys in the form of a stapled set comprising of a cover letter explaining the circumstances and purpose of the stilt; a survey with socio-demographic data; a questionnaire on the pelvic floor ailments; and a pen were distributed either in a white envelope marked with a stamp and a return address or in a white unmarked envelope when it was possible to return the surveys to the survey collection container. Physically active women were tested in the same way in fitness centers, gyms, swimming pools or recreational areas in parks in Upper Silesia. Women not physically active were selected from the civil servants in the voivodeship of Silesia. 

The survey did not specify the time required to complete the questionnaire; it was possible to return the surveys back on the same day or the next day or send them back by post.

The Australian Pelvic Floor Questionnaire (APFQ) [16] was used to measure the extent of pelvic floor dysfunctions in women from all of the three groups. Authors of the APFQ questionnaire intended for it to be completed by the respondents themselves, and it covers all of the areas of pelvic floor dysfunctions such as bladder function, bowel function, prolapse of reproductive organs and sexual function. It measures the symptoms of dysfunctions in the four listed areas as well as their severity, impact on the quality of life and nuisance by applying a 4-point scale for each symptom. On a 4-point scale, 0 means no problem at all and 3 indicates a great problem. The 4-point scale was not used for the frequency of bowel movement, consistency of stool, degree of vaginal lubrication or reason for sexual abstinence [16]. The questionnaire section regarding bladder function contains 15 questions (maximum number of points: 45), the bowel function section contains 12 questions (maximum number of points: 34), the pelvic organ prolapse section has 5 questions (maximum number of points: 15), and the sexual function section contains 10 questions (maximum number of points: 21). The total number of points for all of the four areas of the pelvic floor dysfunction is a maximum of 40 points and it is the sum of the results for each area. The score for the particular area is calculated as the sum of the points in the section of interest, divided by the maximum number of points that can be obtained in the area of pelvic floor dysfunction multiplied by 10. In each of the 4 measured areas of pelvic floor dysfunction, it was possible to obtain from 0 to 10 points [17,18]. The APFQ questionnaire has been validated several times, which confirms its reliability [17,18,19,20], as well as a strong and significant correlation with other validated questionnaires [21].

The surveyed women completed the questionnaire anonymously and on their own. The study was authorized by the Bioethics Committee for Scientific Studies at the Jerzy Kukuczka Academy of Physical Education of Katowice. All of the study procedures were performed according to the Helsinki Declaration of Human Rights of 1975 (modified in 1983).

### 2.3. Statistical Analysis

The characteristics of the participants were described by mean and standard deviation. The occurrence of pelvic floor dysfunction symptoms is presented as a percentage within the groups. Differences in the demographic parameters (age, height, weight, BMI and childbirth delivery method) and the APFQ scores were analyzed by a one-way analysis of variance (ANOVA) with the “group” as a between-subjects factor. For significant results, the post hoc Tukey’s analysis was performed. Differences in the number of childbirths and the frequency of the symptoms were analyzed by the Chi^2^ test. The level of significance was set at *p* < 0.05.

## 3. Results

Table 1 shows demographic data of the groups surveyed. The statistical analysis did not reveal statistically significant differences in demographic data or in the number of childbirth deliveries, drinking of tea, coffee and alcohol and smoking of cigarettes. There were statistical differences noted in drinking water between the analyzed groups. However, lower values of drinking water in the WNPA group (Tukey’s test vs. WPHR group *p* = 0.025 and PAW group *p* = 0.016) did not have significant impact on the values of the individual indicators of the pelvic floor dysfunction symptoms (coefficient of Pearson’s test were 0.126 for Bladder and Bowles scores, −0.162 for POP score and −0.073 for sex score; at all times *p* > 0.05). 

Table 2 lists the rate of bladder and bowel functions, prolapse of reproductive organs, sexual function as well as an overall rate of pelvic floor dysfunction. With the exception of the sexual function, the lowest average rate values were found in women practicing recreational horseback riding, and the highest in the group of women not physically active. The overall pelvic floor dysfunction rate in the WPHR group was only slightly lower for women in the PAW group (by 0.06 points on average) and by 0.9 points on average compared to women in the WNPA group. Statistically significant intergroup differences were recorded only for the bowel function rate (*p* = 0.021). The post-hoc analysis (Tukey’s test) confirmed significantly lower values of this rate in the WPHR group (*p* = 0.033) and in the PAW group (*p* = 0.05) as compared to the WNPA group. 

### 3.1. Bladder Symptoms

Most of the surveyed women claimed that they did not have any negative symptoms of urinary bladder or lower urinary tract function (except for the urgency symptom in the WNPA group). The surveyed women who complained of such symptoms, usually reported that these symptoms were present occasionally (score—1 point). In individual cases, some women claimed that the symptom occurred always/daily or more than 15 times (for urinary frequency) and more than 3 times (for nocturia symptoms), which were rated at 3 points. In the WPHR group, this was the case for urinary frequency (in three women), in the case of urgency (one woman), in the case of a weak stream (one woman), in the sense of incomplete bladder emptying (one woman), and in the case of dysuria (one woman). In addition, in individual cases, the highest severity of the symptoms (3 points) was reported by women in the PAW group (one case for the following symptoms: urinary frequency, nocturia, urgency). A similar situation was observed by the members of WNPA group, where one person reported the maximum severity with regard to the following symptoms: nocturia, stress incontinence, incomplete bladder emptying, and dysuria. For most of the symptoms rated in this part of the questionnaire, the women practicing horseback riding recreationally reported the symptoms of urinary bladder and lower urinary tract dysfunctions the least frequently. On the contrary, the women not physically active usually indicated the occurrence of such symptoms the most often. However, the Chi^2^ test did not reveal any statistically significant differences (only if stress incontinence was present, the differences to the disadvantage of both control groups were close to the level of *p* = 0.061). (Figure 2).

### 3.2. Bowel Symptoms

It was only in the assessment of the bowel function that a symptom was recorded that occurred in the majority of the surveyed women within all of the groups. The occurrence of defecation straining was reported by 76.08 of the women in the WPHR group, 72.34% of the women in the PAW group, and 89.36% of the women in the WNPA group. The vast majority of these women reported occasional (<1/week) occurrence of this symptom (67.39% in the WNPA group, 70.21% in the PAW group, and 76.59% in the WNPA group). Only two women not physically active reported a daily (3 points) occurrence of this symptom. With regard to other bowel symptoms, it was only among the women in the WNPA group where the majority reported having such problems. These symptoms were related to the bowel consistency symptom (72.34%) and the occurrence of fecal urgency (51.06%). No women in the WPHR group reported the maximum severity of the bowel symptoms (score—3 points). Among the women in the PAW group, this applied to one case (incomplete bowel movement). Most often, the highest degree of severity was recorded among the members of the WNPA group (defecation straining, two cases; flatus incontinence, four cases; fecal urgency, one case). The Chi^2^ test showed that problems with bowel consistency were statistically significant more frequently in the women in the WNPA group as compared to the women in the WHR and PAW groups (*p* = 0.0045). At a level close to the statistical significance (*p* = 0.052), there were differences in the occurrence of obstructed defecation between the members of the PAW group and the WPHR group. (Figure 3).

### 3.3. Pelvic Organ Prolapse Symptoms

Amongst all of the symptoms related to the pelvic floor dysfunction, surveyed women indicated that the occurrence of prolapse symptoms was the least frequent. In both, the WPHR and PAW groups, symptoms of this type occurred in five cases (prolapse sensation, 1 case in each group; vaginal pressure or heaviness, three cases in each group; prolapse reduction to void, one case in each group), and in each case the symptoms were sporadic (APFQ score—1 point). In the WNPA group, these types of symptoms occurred in six women (prolapse sensation, two women; vaginal pressure or heaviness, four women), and were occasional (<1/week). (Figure 4).

### 3.4. Sex Dysfunction Symptom

The same number of women (5) in each of the groups were not sexually active. A similar percentage of women was sexually active most days or daily (60.86% in the WPHR group, 59.57% in the PAW group and 55.32% in the WNPA group). Sexual function disorders were found in a small proportion of the respondents, with the exception of dyspareunia symptoms in the WNPA group being present in 40.42% of the women. Nobody in the WPHR or PAW groups reported a high severity of the symptoms (APFQ score —3). On the other hand, in the WNPA group, this was indicated by six women (non vaginal sensation during intercourse, three women; vaginal laxity always, one woman; vaginal tightness always, one woman; dyspareunia always, one woman). Only for occasional coital incontinence symptoms (five cases in the PAW group, with the absence of such cases in the WPHA and in the WNPA group), the Chi^2^ test showed statistically significant differences (*p* = 0.0059). (Figure 5).

## 4. Discussion

Horse riding is a physical activity that significantly engages not only the pelvis but also pelvic floor itself. However, the survey results show that recreational horseback riding does not affect the subjective sensation of the pelvic floor adversely.

The assessment of the pelvic floor was applied to bladder function, bowel function, prolapse of reproductive organs, and sexual function.

The movement of a horse in each gait vast majority of the time is carried out in the vertical axis (up and down), with this movement being the most clearly seen in the canter. In the trot the movement amplitude is lower, but the frequency is higher, which makes it difficult for the rider’s joints to absorb the moves in a smooth way. Jumps, as a part of physical activity, are identified by the researchers as a risk factor that may contribute to urinary incontinence in women [22,23]. The landing phase after the hop is considered a risk factor. According to Hagovska et al. [22], volleyball players have a 116% higher chance of getting stress urinary incontinence compared to women who play other types of sports. Jumps in horseback riding are also present. Nonetheless, they are different types of jumps. The horse’s jump in the canter is its natural action in which the landing phase is also the preparation phase for the next jump. This activity is very well cushioned by the horse’s joints. A skilled rider (and such were the surveyed women) is able to move their pelvis and trunk smoothly in line with the movement of the horse’s back. Unlike landing following a jump in sports such as volleyball, there is no sudden impact on the ground in the case of horseback riding, which, simply speaking, would cause a rapid pressure of the abdominal cavity and pelvis major on the pelvic floor. In other words, a type of a rapid impact occurs on the structure of the bottom of the pelvis, in a form of a pull mechanism. During horseback riding, the pelvis makes a smooth rocking movement with varying amplitudes that are related for example to the horse’s gait rate getting adapted to the terrain configuration. This causes changes in the rate and amplitude of the female pelvis movement, which is beneficial to the woman. Therefore, such effort does not strain the pelvic structures. This proves, and is also reflected by the survey results, that stress urinary incontinence among female horse riders is a much slighter problem as compared to physically active women and women not physically active. Thus, it seems that the rider’s movement in a vertical axis does not overload the pelvic floor in such a way as jumps during a volleyball game or a CrossFit workout. On the other hand, it should be noted that the pelvic floor of a woman riding a horse frequently undergoes a high frequency of tossing through the horse’s back. However, according to Clayton and Hoobs [10], the absorption of the horse’s movement by the rider’s hock joints consists of rhythmic deepening of their dorsiflexion, with the heel remaining below the toes [24]. The position of the hock joints of the surveyed women during horseback riding may be crucial for bladder function, as according to Lee [25], the foot dorsiflexion (typical for horseback riding) is accompanied by the activation of an anterior tilt of the pelvis and pelvic floor muscles, which leads to an increased pelvic floor tension. Whereas in the plantarflexion, adopted by a person performing a jump, there is no movement of the pelvis and the tension of the pelvic floor is significantly lower. As a result, there are rhythmic changes in the pelvic floor tension that prevent overloading. 

Bowel function of women practicing horseback riding and physically active women is very similar, whereas a greater degree of constipation is observed in the group of women not physically active. According to the studies on constipation in the group of 62,032 women aged 36–61, the occurrence of constipation depends on the level of physical activity and the diet maintained, in particular on the fiber content in the diet [26]. Studies on eating habits among Polish women practicing fitness show that physical activity has positive effects on their eating habits, which were generally assessed as sufficiently good [27]. On the other hand, an experiment excluding the effects of the diet showed that shortly after the physical activity is ceased, there is a problem of constipation and a marker of inflammation in the bowels occurs which cannot be attributed to changes in the microbiota [28]. In the light of the presented studies, it appears that the difference in the intensity of constipation among the groups of the surveyed women was directly due to the level of physical activity in which recreational horseback riding gives similar positive effects as other forms of physical activity. 

The results of the studies showed that the prolapse of pelvic organs was not a problem in any of the surveyed groups.

In regard to sexual function, the women practicing horseback riding rated this function as slightly better than the women from the reference groups. However, this was not confirmed statistically. These results confirm the observations indicating that horseback riding does not have a negative impact on this area of life for female recreational horseback riders. This is in line with the results obtained by Alanee et al. [15].

The results of this cross-sectional study indicate that recreational horseback riding performed by women does not result in increased symptoms of pelvic floor dysfunction compared with control groups. It is unknown whether such a risk could occur in women who undertake this type of activity later in life (e.g., in middle age women). Therefore, further study should be undertaken to assess the influence of horseback riding on the pelvic floor in the period around and after the menopause, which is critical for the appearance of dysfunction in this part of the body. It is also unknown what effect different riding techniques exert on the activity of the pelvic floor. In future studies, the pre-activity and reflex activity of pelvic floor muscles during horseback riding should therefore be analyzed, both in women with a properly functioning pelvic floor and in women with symptoms of pelvic floor dysfunction. Such study would allow to assess the possible therapeutic potential of horseback riding in relation to pelvic floor dysfunction. 

## 5. Conclusions

The results of the study show that women practicing horseback riding are not suffering from severe lower urinary tract problems and that there is no negative impact of recreational equestrianism on bowel function. Moreover, this study has demonstrated that there are no severe symptoms related to the prolapse of pelvic organs and no adverse effects on sexual function among women practicing recreational horseback riding were found. Therefore, it can be concluded that compared to women practicing other types of physical activity and physically inactive women, horseback riding does not contribute to the appearance of symptoms of pelvic floor dysfunction in women.

## Figures and Tables

**Figure 1 ijerph-19-02108-f001:**
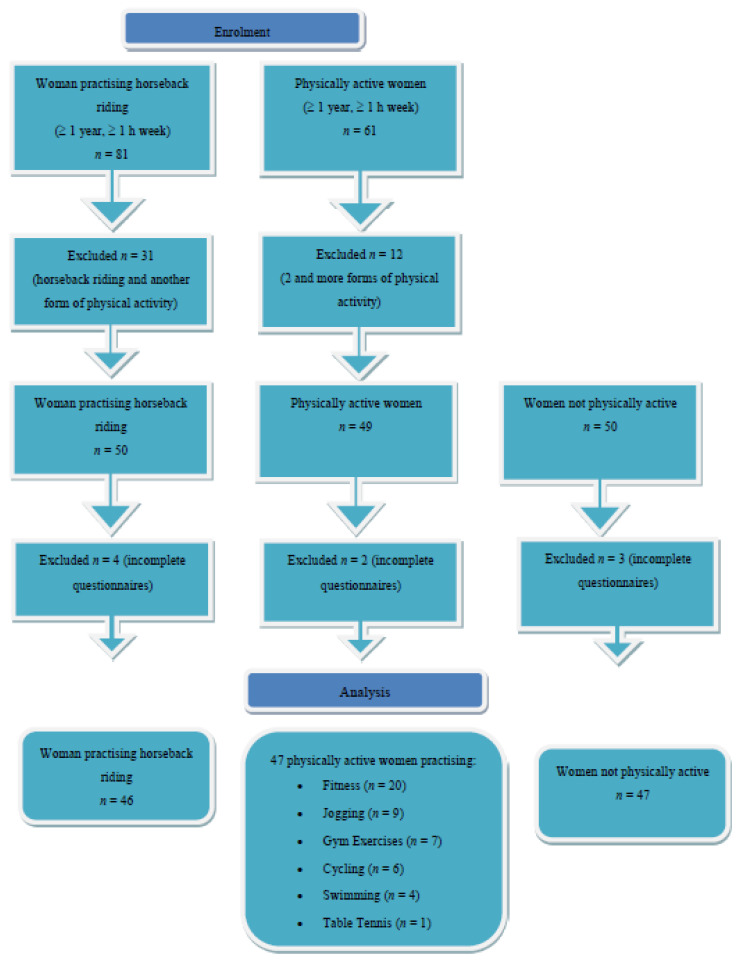
Flow diagram of the study phases.

**Figure 2 ijerph-19-02108-f002:**
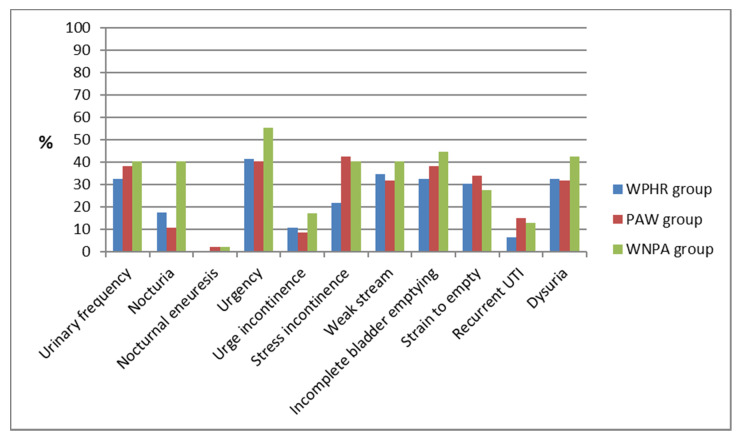
Occurrence of bladder symptoms by groups.

**Figure 3 ijerph-19-02108-f003:**
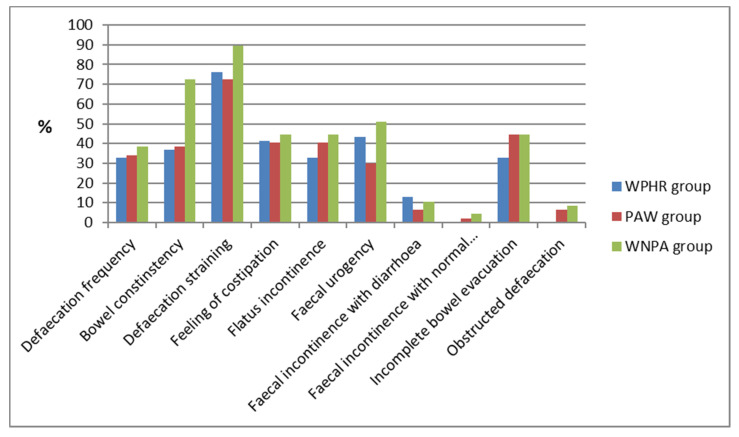
Occurrence of bowel symptoms by groups.

**Figure 4 ijerph-19-02108-f004:**
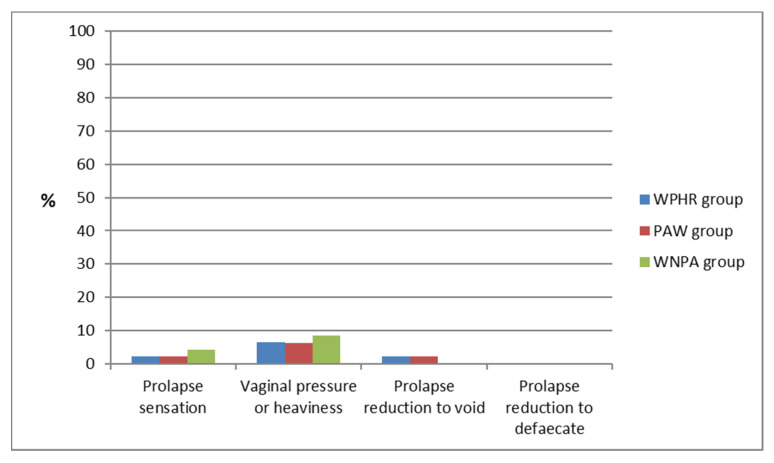
Occurrence of prolapse symptoms by groups.

**Figure 5 ijerph-19-02108-f005:**
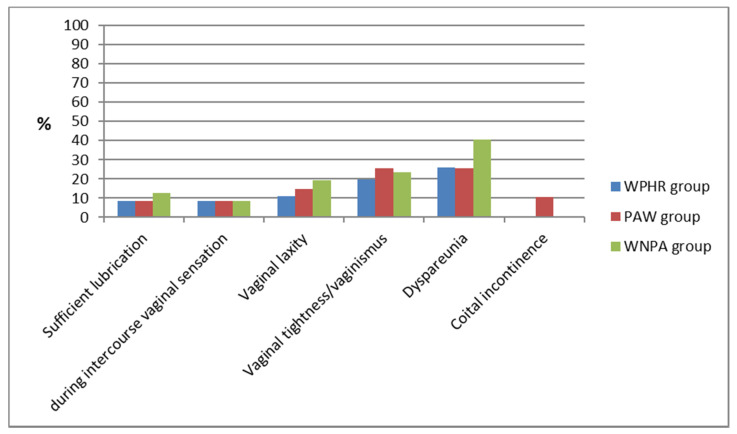
Occurrence of sexual dysfunction symptoms by groups.

**Table 1 ijerph-19-02108-t001:** Demographic data of the participants.

Characteristics	WPHR Group (n = 46)	PAW Group (n = 47)	WNPA Group (n = 47)	*p* Value
Age (years)	33.37 (10.2)	35.94 (9.9)	36.85 (11.2)	0.255 ^a^
Weight (kg)	68.9 (19.8)	65.1 (11.8)	66.3 (13.8)	0.490 ^a^
Height (cm)	167.4 (5.9)	165.9 (4.7)	166.0 (4.9)	0.314 ^a^
BMI	24.47 (6.1)	23.93 (4.0)	24.06 (5.1)	0.880 ^a^
Childbirth natural (n) caesarean section (n) average for woman	18 13 0.72 (0.8)	24 14 0.85 (0.9)	33 14 1.0 (0.9)	0.420 ^b^ 0.668 ^b^ 0.214 ^a^
Drinking water (n of bottles × 0.5l)	3.22 (1.2)	3.26 (1.0)	2.55 (1.2)	0.008 ^a^
Drinking tea (n of cups)	1.89 (0.8)	1.8 (1.2)	2.0 (0.9)	0.642 ^a^
Drinking coffee (n of cups)	1.52 (1.0)	1.52 (0.9)	1.68 (0,8)	0.659 ^a^
Alcohol	Yes 31 No 15	Yes 29 No 18	Yes 23 No 24	0.168 ^b^
Smoking	Yes 4 No 42	Yes 11 No 36	Yes 5 No 42	0.115 ^b^

^a^ ANOVA; ^b^ Chi^2^.

**Table 2 ijerph-19-02108-t002:** Comparison of scale scores between the groups.

Score	WPHR Group	PAW Group	WNPA Group	*p* Value ^a^
Bladder	0.88 (0.8) 0.61–1.15	0.96 (0.9) 0.7–1.21	1.16 (0.9) 0.87–1.44	0.317
Bowel	1.07 (0.8) 0.82–1.32	1.1 (0.8) 0.86–1.34	1.52 (0.9) 1.24–1.8	0.021 *
POP	0.2 (0.4) 0.07–0.33	0.21 (0.5) 0.07–0.35	0.21 (0.4) 0.08–0.35	0.993
Sex	0.74 (1.1) 0.4–1.07	0.67 (0.9) 0.39–0.95	0.89 (1.3) 0.49–1.3	0.639
Total	2.88 (2.4) 2.15–3.62	2.94 (2.1) 2.34–3.54	3.78 (2.6) 3.0–4.56	0.131

POP, pelvic organ prolapse; ^a^ ANOVA; * Statistically significant difference between the groups.

## Data Availability

The data that support the findings of this study are available from the corresponding author, upon reasonable request.

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
