# Peer review of "Self-Assessment of the Pelvic Floor by Women Practicing Recreational Horseback Riding"

_ijerph, 2022, doi:10.3390/ijerph19042108_

Round 1

Reviewer 1 Report

The idea of this study seems to be very interesting and necessary since horse riding has always been considered as a risk factor for pelvic floor dysfunctions. Nevertheless, in my opinion, the authors should re-write the study taking in account important issues:

INTRODUCTION: The introduction is large and contains information which does not really fit the subject of the study. In addition, there are only two references of real antecedents about the effects of horse riding in pelvic floor dysfunctions. I encourage them to summarize the first part of the introduction (until line 70) and focus on the subject giving more previous evidences.

CONCLUSIONS: This a cross-sectional study, and mainly the statistical analysis has beed purely descriptive, so the conclusions given by the authors, according to which, horse riding cannot be considered as a risk factor, are not coherent with the study design.

Author Response

  1. Introduction section has been shortened. The content that is less relevant to the research problem has been removed from this section (text placed between the lines: 44-57 and 61-66).
  2. The content of the conclusion was rewritten as suggested by the reviewer.

Reviewer 2 Report

This is a very interesting article. It needs gramatical and English revision. I would consider exclude the ones 46-49. In the introduction try to vary the references. The authors used reference 10 for a large part of the introduction. And they made the last discussion paragraph as conclusion and  included a conclusion item in the end. I suggest to resume the conclusion just in the conclusion item.  

Author Response

  1. As suggested, the section of text from introduction in the lines 46-49 has been removed.
  2. As per the reviewer’s suggestions, reference 10 was amended and used specifically in relation to the rider’s pelvic movements in each type of horse gait.
  3. The last part of the discussion has been redrafted so that it does not duplicate the content from the conclusion.

Reviewer 3 Report

Thank you for giving me the possibility to review the paper "Self-assessment of the Pelvic Floor by Women Practising Recreational Horseback Riding". The present paper aimed to compare the condition of the pelvic floor in women who are involved in regular recreational horseback riding with both physically active women as well as women not undertaking any recreational physical activity. For this purpose, 140 potentially healthy women aged 17 to 61 were included in the present study and divided into the following groups: women practising horseback riding (WPHR), physically active women (PAW)  and women not physically active (WNPA).

The present concerns should be addressed before considering the present paper for publication on IJERPH:

  • the age interval of the recruited women is too large; please consider to reduce the age interval or perform a more detailed subgroups analysis:
  • please compare pre-menopausal women to post-menopausal women;
  •  please improve the discussion section by focusing on the relevance of the findings of the present paper.

Author Response

  1. We appreciate the reviewer’s comment related to age interval. However, the reviewer’s suggestion regarding the analysis within subgroups (pre- and post-menopausal women) in the case of the research material presented in this paper is not applicable due to the number of the women surveyed. In the nearest future there are plans for the research that intends to include a greater percentage of women in the perimenopausal age.
  2. Minor changes to discussion section of the paper were implemented regarding the results of the study.

Reviewer 4 Report

Interesting study, timely with regard to new interest in horseback riding as form of exercise (and particularly due to outdoor activities during COVID-19 pandemic), and particularly in women, therefore of interest in terms of potential effects on pelvic floor dysfunction and other endpoints, overall well presented with appropriately matched cohorts, of interest in terms of initial reassuring findings with regard to horseback riding in this population. Ideally would be higher enrollment/higher power of study, of interest although somewhat for a specific niche of patients.

Author Response

Some moderate changes to the language and style were made.

Round 2

Reviewer 1 Report

After review the second draft provided by the authors, and analysisng their answer to my prior requests, I consider the manuscript has been now improved enough to be published.

But, Figure 1 (Figure 1. Flow diagram of the study phases.) needs to be better fitted in one page. It is uncomfortable to read between two pages.

Author Response

Thank you for the suggestion regarding Figure 1. We have formatted this Figure and it is now fitted in one page. Thank you for advising that the paper is now suitable for publication. 

Reviewer 3 Report

Thankyou for giving me the possibility to review the revised version of the paper. All the raised concerns have been addressed by authors, therefore the paper is now suitable of publication. 

Author Response

Thank you for the review and for advising that the paper is now suitable for publication.